# Defocus Corrected Large Area Cryo-EM (DeCo-LACE) for label-free detection of molecules across entire cell sections

**Johannes Elferich[1]\*, Giulia Schiroli[2,3,4], David T Scadden[2,3,4], Nikolaus Grigorieff[1]\***

[1]RNA Therapeutics Institute, University of Massachusetts Chan Medical School, Howard Hughes Medical Institute, Worcester, United States; [2]Department of Stem Cell and Regenerative Biology, Harvard University, Cambridge, United States; [3]Harvard Stem Cell Institute, Cambridge, United States; [4]Center for Regenerative Medicine, Massachusetts General Hospital, Boston, United States

**Abstract** A major goal of biological imaging is localization of biomolecules inside a cell. Fluorescence microscopy can localize biomolecules inside whole cells and tissues, but its ability to count biomolecules and accuracy of the spatial coordinates is limited by the wavelength of visible light. Cryo-electron microscopy (cryo-EM) provides highly accurate position and orientation information of biomolecules but is often confined to small fields of view inside a cell, limiting biological context. In this study, we use a new data-acquisition scheme called Defocus-Corrected Large-Area cryo-EM (DeCo-LACE) to collect high-resolution images of entire sections (100- to 250-nm-thick lamellae) of neutrophil-like mouse cells, representing 1–2% of the total cellular volume. We use 2D template matching (2DTM) to determine localization and orientation of the large ribosomal subunit in these sections. These data provide maps of ribosomes across entire sections of mammalian cells. This high-throughput cryo-EM data collection approach together with 2DTM will advance visual proteomics and provide biological insight that cannot be obtained by other methods.

**\*For correspondence:**
Johannes.Elferich@umassmed.edu (JE);
niko@grigorieff.org (NG)

## Editor's evaluation

The work details a valuable method of defocus corrected large area cryo-EM (DeCo-LACE). The data-acquisition approach is highly complementary to the research group's previous work of using high-resolution 2D template-matching (2DTM) to identify macromolecular complexes in dense and heterogeneous cellular specimens. Overall, the data presented shows that DeCo-LACE is a solid approach to locating large ribosomal subunits in FIB-lamella at scale.

## Introduction

A major goal in understanding cellular processes is the knowledge of the amounts, location, interactions, and conformations of biomolecules inside the cell. This knowledge can be obtained by approaches broadly divided into label- and label-free techniques. In label-dependent techniques a probe is physically attached to a molecule of interest that is able to be detected by its strong signal, such as a fluorescent molecule. In label-free techniques, the physical properties of molecules themselves are used for detection. An example for this is proteomics using mass-spectrometry (*Griffin et al., 2010*). The advantage of label-free techniques is that they can provide information over thousands of molecules, while label-dependent techniques offer highly specific information for a few molecules. In particular, spatial information is primarily achieved using label-dependent techniques, such as fluorescence microscopy (*Lichtman and Conchello, 2005*).

Cryo-electron microscopy (cryo-EM) has the potential to directly visualize the arrangement of atoms that compose biomolecules inside of cells, thereby allowing label-free detection with high spatial accuracy. This has been called visual proteomics (*Nickell et al., 2006*). Since cryo-EM requires thin samples (<500 nm), imaging of biomolecules inside cells is restricted to small organisms, thin regions of cells, or samples that have been suitably thinned. Thinning can be achieved either by mechanical sectioning (*McDowall et al., 1983*) or by milling using a focused ion beam (FIB) (*Villa et al., 2013*). This complex workflow leads to a low throughput of cryo-EM imaging of cells and is further limited by the fact that at the required magnifications, typical fields of view (FOV) are very small compared to mammalian cells, and the FOV achieved by label-dependent techniques such as fluorescence light microscopy. The predominant cryo-EM technique for the localization of biomolecules of defined size and shape inside cells is cryo-electron tomography (*Gan and Jensen, 2012*). However, the requirement of a tilt series at every imaged location and subsequent image alignment, severely limits the throughput for molecular localization.

An alternative approach is to identify molecules by their structural fingerprint in single projections using 2D template-matching (2DTM) (*Rickgauer et al., 2017*; *Rickgauer et al., 2020*; *Lucas et al., 2021*). In this method, a 3D model of a biomolecule is used as a template to find 2D projections that match the molecules visible in the electron micrographs. This method requires a projection search on a fine angular grid, and the projections are used to find local cross-correlation peaks with the micrograph. Since the location of a biomolecule in the z-direction causes predictable aberrations to the projection image, this method can be used to calculate complete 3D coordinates and orientations of a biomolecule in a cellular sample (*Rickgauer et al., 2020*).

Here we apply 2DTM of the ribosome large subunit (LSU) to a conditionally immortalized *Mus musculus* (mouse) cell line that gives rise to functional mature neutrophils (*Sykes et al., 2016*). We chose these cells because genetic defects in the ribosome machinery often lead to hematopoietic disease (*Kampen et al., 2020*) and direct quantification of ribosome location, number and conformational states in hematopoietic cells could lead to new insight into hematopoietic disease (*Dolezal et al., 2018*). To increase the amount of collected data and to provide unbiased sampling of the whole lamella, we devised a new data-acquisition scheme, Defocus-Corrected Large Area Cryo-Electron microscopy (DeCo-LACE). 2DTM allows us to test whether aberrations caused by large beam-image shifts and highly condensed beams deteriorate the high-resolution signal. We find that these aberrations do not impede LSU detection by 2DTM. The resulting data provide a description of ribosome distribution in an entire lamella, which represent 1–2% of the cellular volume. We find a highly heterogeneous density of ribosomes within the cell. Analysis of the throughput in this method suggests that for the foreseeable future computation will be the bottleneck for visual proteomics.

## Results

### 2DTM detects large ribosomal subunits in cryo-FIB lamellae of mammalian cells

FIB-milled *Saccharomyces cerevisiae* (yeast) cells are sufficiently well preserved to permit localization of 60S ribosomal subunits with 2DTM (*Lucas et al., 2022*). Due to the larger size of mammalian cells compared to yeast cells, it was unclear whether plunge freezing would be adequate to produce vitreous ice across the whole volume of the cell. To test this we prepared cryo-lamellae of mouse neutrophil cells. A low magnification image of a representative lamella clearly shows cellular features consistent with a neutrophile-like phenotype, mainly a segmented nucleus and a plethora of membrane-organelles, corresponding to the granules and secretory vesicles of neutrophils (*Figure 1A*). We then proceeded to acquire micrographs on this lamella with a defocus of 0.5–1.0 μm, 30 e⁻/Å²/s exposure and 1.76 Å pixel size. We manually selected multiple locations in the lamella and acquired micrographs using standard low-dose techniques where focusing is performed on a sacrificial area. The resulting micrographs showed smooth bilayered membranes and no signs of crystalline ice (*Figure 1C and D*), indicating successful vitrification throughout the lamella.

We used an atomic model of the 60S mouse ribosomal subunit (6SWA) for 2DTM (*Kraushar et al., 2021*). In a subset of images, the distribution of cross-correlation scores significantly exceeded the distribution expected from images devoid of detectable targets. In the resulting scaled maximum-intensity projections (MIPs), clear peaks with SNR values up to 10 were apparent (*Figure 1—figure*

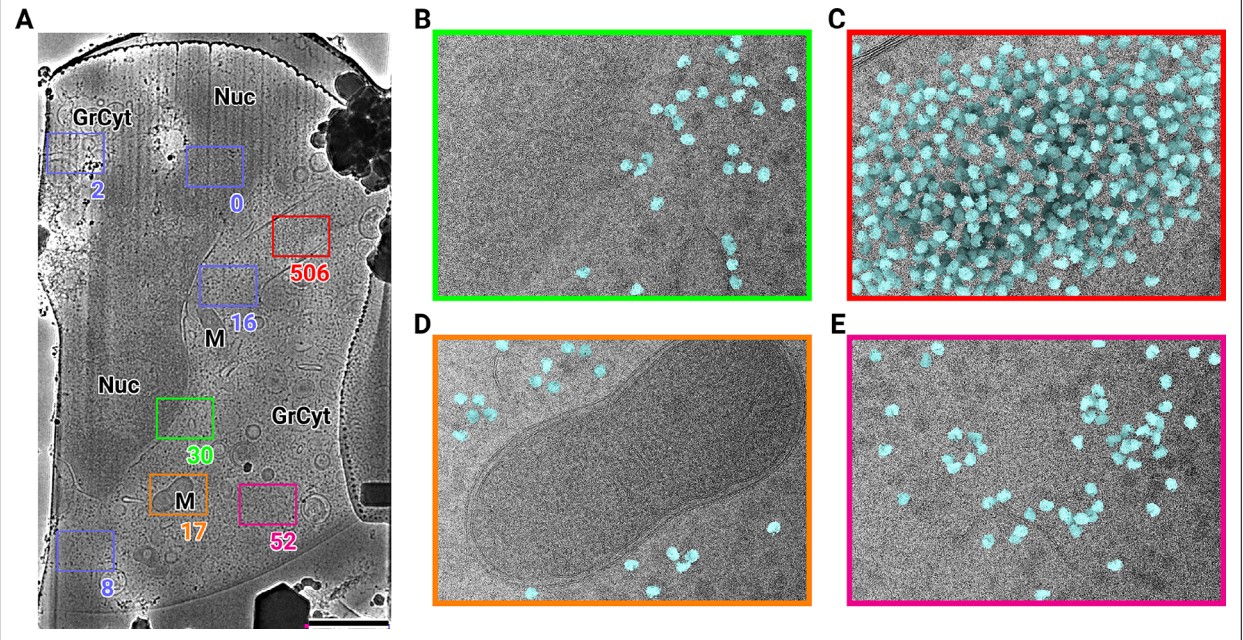

**Figure 1.** 2D template matching of the large subunit of the ribosome in fib-milled neutrophil-like cells (**A**) Overview image of the lamella. Major cellular regions are labeled, as Nucleus (Nuc), Mitochondria (M), and granular cytoplasm (GrCyt). FOVs where high-magnification images for template matching where acquired are indicated as boxes with the number of detected targets indicated on the bottom right. FOVs displayed in Panels B-E are color-coded. Scalebar corresponds to 1 μm. (**B–E**) FOVs with projection of detected LSUs shown in cyan. (**B**) Perinuclear region, the only detected targets are in the cytoplasmic half. (**C**) Cytoplasmic region with high density of ribosomes (**D**) Mitochondrium, as expected there are only detected LSUs in the cytoplasmic region (**E**) Cytoplasm, with low density of ribosomes.

The online version of this article includes the following figure supplement(s) for figure 1:

**Figure supplement 1.** 2D template matching of the large subunit of the ribosome in fib-milled neutrophil-like cells.

*supplement 1A*). Using a threshold criterion to select significant targets (see Methods), we found that in images of cytosolic compartments there were 10–500 ribosomes within one micrograph (*Figure 1B–E*). Notably, we found no targets in areas corresponding to the nucleus (*Figure 1B*) or mitochondria (*Figure 1D*). In the cytoplasm, we found a highly variable number of targets, only ~50 in some exposures (*Figure 1E*) and up to 500 in others (*Figure 1C*). However, it is unclear whether this ten-fold difference in local ribosome concentration is due to technical variation, such as sample thickness, or biological variation. To differentiate between the two we reasoned it was important to not manually choose imaging regions and to collect larger amounts of data. We therefore set out to collect cryo-EM data for 2DTM from mammalian cell lamellae in a high-throughput unbiased fashion.

## DeCo-LACE for 2D imaging of whole lamellae

In order to obtain high-resolution data from complete lamellae, we developed a new approach for data collection. This approach uses three key strategies: (1) every electron that exposes a fresh area of the sample is collected on the camera, (2) image shift is used to precisely and quickly raster the surface of a lamella, and (3) focusing is done without using a sacrificial area (*Figure 2A and B*).

To ensure that every electron exposing a fresh area of the sample is captured by the detector, we adjusted the electron beam size to be entirely contained by the detector area. During canonical low-dose imaging, the microscope is configured so that the focal plane is identical to the eucentric plane of the specimen stage. This leaves the C2 aperture out of focus, resulting in ripples at the edge of the beam (*Figure 2D*). While these ripples are low-resolution features that likely do not interfere with 2DTM (*Rickgauer et al., 2017*), we also tested data collection under conditions where the C2 aperture is in focus (fringe-free, *Figure 2E*; *Konings et al., 2019*).

We then centered a lamella on the optical axis of the microscope and used the image shift controls of the microscope to systematically scan the whole surface of the lamella in a hexagonal pattern (*Figure 2A and C*). Instead of focusing on a sacrificial area, we determined the defocus from every

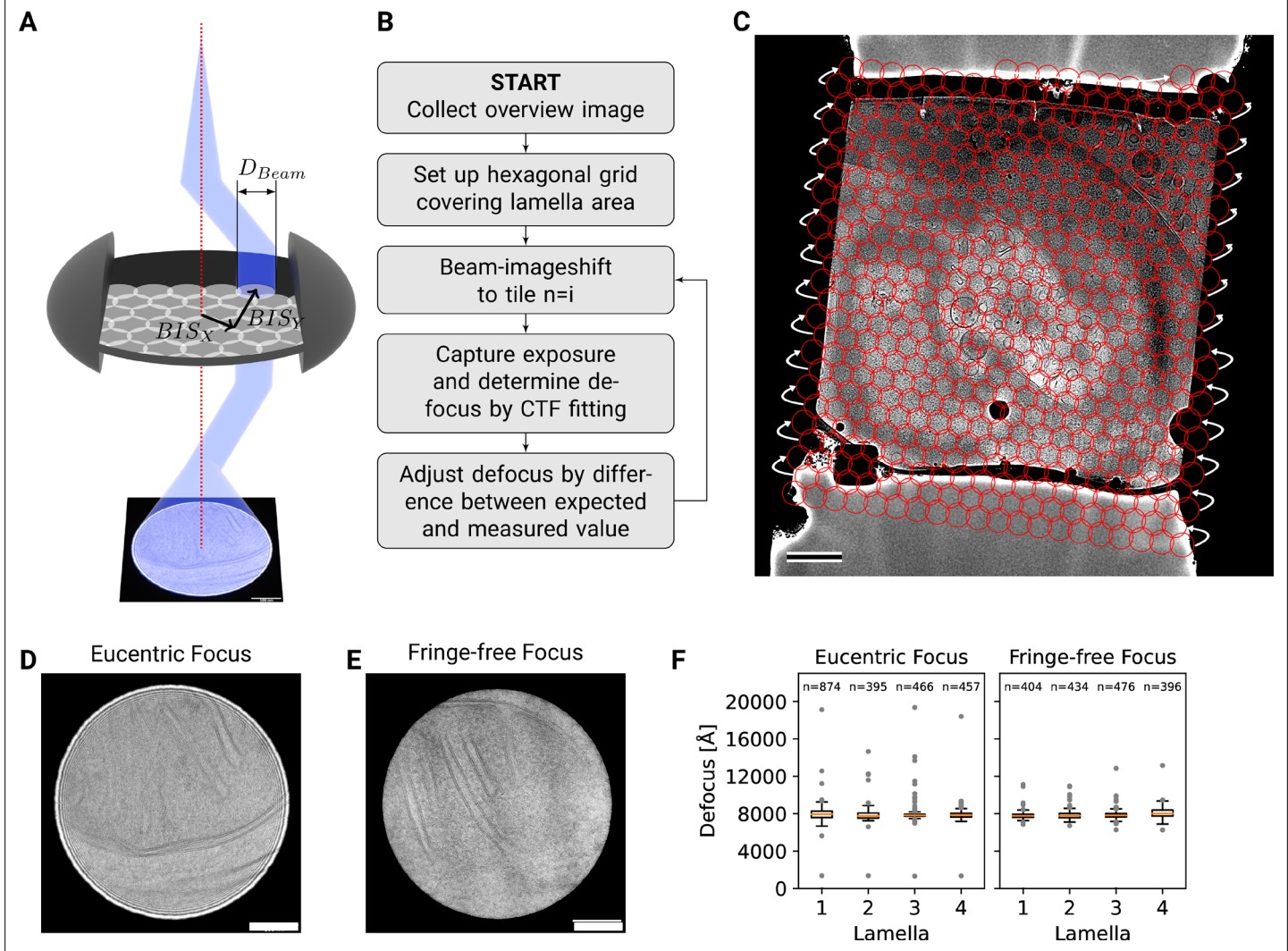

**Figure 2.** DeCo-LACE approach. (**A**) Graphic demonstrating the data-collection strategy for DeCo-LACE. The electron beam is condensed to a diameter $D_{Beam}$ that allows capturing of the whole illuminated area on the camera. Beam-image shift along X and Y ($BIS_X$,$BIS_Y$) is used to scan the whole lamella. (**B**) Diagram of the collection algorithm. (**C**) Example overview image of a lamella with the designated acquisition positions and the used beam diameter indicated with red circles. Scalebar corresponds to 1 μm. (**D**+**E**) Representative micrographs taken with a condensed beam at eucentric focus (panel D) or fringe-free focus (panel E). Scalebar corresponds to 100 nm. (**F**) Boxplot of defocus measured by ctffind of micrographs taken by the DeCo-LACE approach on four lamellae imaged at eucentric focus and four lamellae imaged with fringe-free focus. Sample size n is indicated for each lamella.

The online version of this article includes the following figure supplement(s) for figure 2:

**Figure supplement 1.** Defocus estimation of individual tiles of DeCo-LACE montages.

exposure after it was taken. The defocus was then adjusted based on the difference between desired and measured defocus (*Figure 2B*). Since we used a serpentine pattern for data collection, every exposure was close to the previous exposure, making large changes in the defocus unlikely. Furthermore, we started our acquisition pattern on the platinum deposition edge to make sure that the initial exposure where the defocus was not yet adjusted did not contain any biologically relevant information.

We used this strategy to collect data on eight lamellae, four using the eucentric focus condition, hereafter referred to as Lamella_EUC, and four using the fringe-free condition, hereafter referred to as Lamella_FFF (*Figure 3A-F*, *Figure 3—figure supplement 4A*). We were able to collect data with a highly consistent defocus of 800 nm (*Figure 2F*), both in the eucentric focus and fringe-free focus condition. To ensure that data were collected consistently, we mapped defocus values as a function of

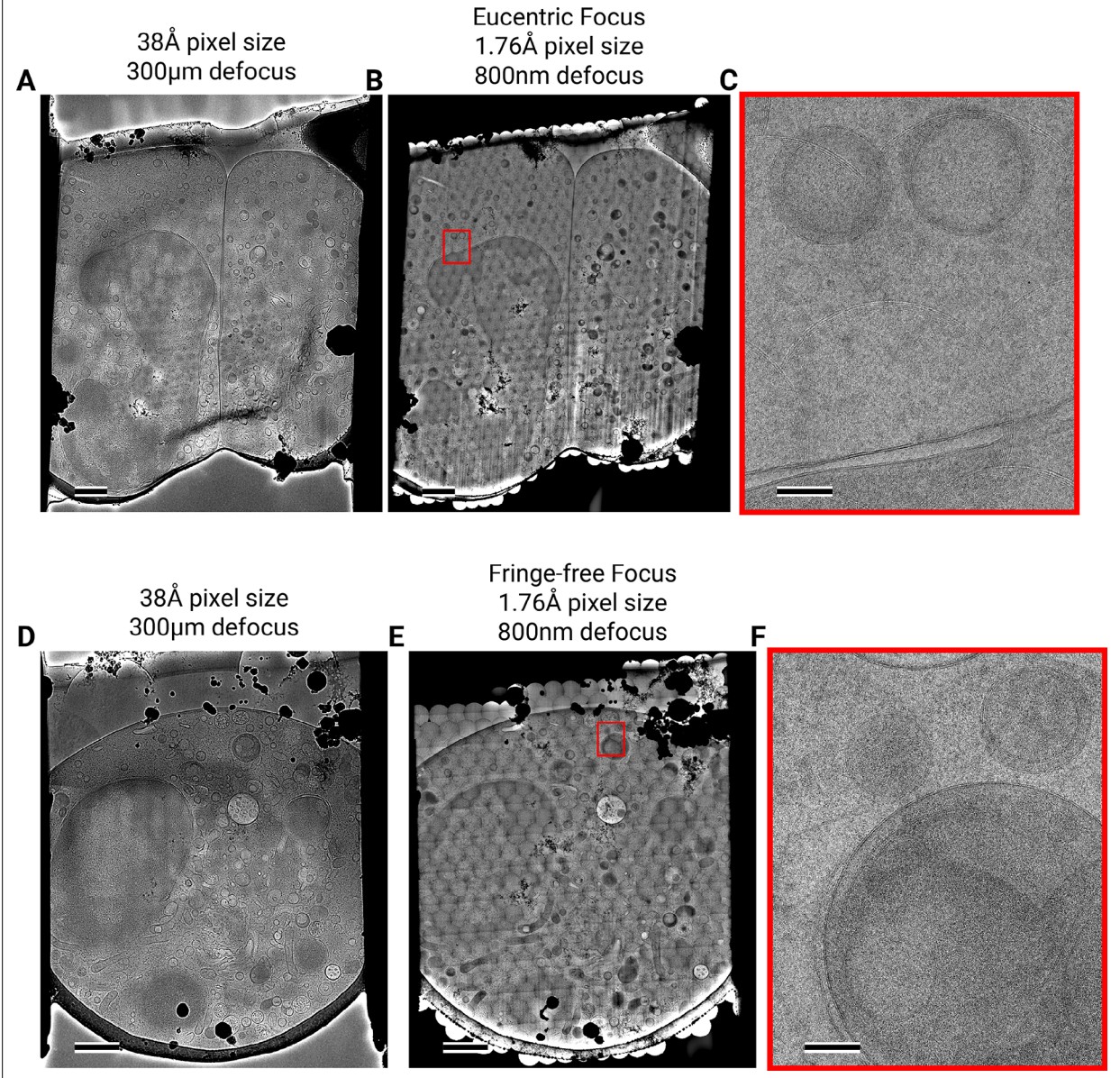

**Figure 3.** Assembling DeCo-LACE exposures into montages. (**A**) Overview image of Lamella_EUC 1 taken at low magnification. Scalebar corresponds to 1 μm. (**B**) Overview of Lamella_EUC 1 created by montaging high magnification images taken with the DeCo-LACE approach. Scalebar corresponds to 1 μm. (**C**) Zoom-in into the red box in panel B. Slight beam-fringe artifacts are visible. Scalebar corresponds to 100 nm. (**D**) Overview image of Lamella_FFF 4 taken at low magnification. Scalebar corresponds to 1 μm. (**E**) Overview of Lamella_FFF 4 created by montaging high magnification images taken with the DeCo-LACE approach. Scalebar corresponds to 1 μm. (**F**) Zoom-in into the red box in panel E. No beam-fringe artifacts are visible. Scalebar corresponds to 100 nm.

The online version of this article includes the following figure supplement(s) for figure 3:

**Figure supplement 1.** Motion correction of movies with condensed beams.

**Figure supplement 2.** Motion correction of individual tiles imaged using the DeCo-LACE approach.

**Figure supplement 3.** Averages of micrographs taken with a condensed beam over vacuum using a Gatan K3 detector.

**Figure supplement 4.** Overview images of lamellae imaged using the DeCo-LACE approach taken at low-magnification.

the applied image shift (*Figure 2—figure supplement 1A*). This demonstrated that the defocus was consistent across a lamella, except for rare outliers and in images containing contamination. We also plotted the measured objective astigmatism of each lamella and found that it varies with the applied image shift, becoming more astigmatic mostly due to image shift in the x direction (*Figure 2—figure*

*supplement 1B*). While approaches exist to correct for this during the data collection (*Wu et al., 2019*), we opted to not use these approaches in our initial experiments. We reasoned that because 2DTM depends on high-resolution information, this would be an excellent test of how much these aberration affect imaging.

We assembled the tile micrographs into a montage using the image-shift values and the SerialEM calibration followed by cross-correlation based refinement (see Methods). In the resulting montages, the same cellular features visible in the overview images are apparent (*Figure 3B+E*, *Figure 3—figure supplement 4B*), however due to the high magnification and low defocus many more details, such as the membrane bilayer separation, can be observed (*Figure 3C+F*). For montages collected using the eucentric condition, there are clearly visible fringes at the edges between the tiles (*Figure 3C*), which are absent in the fringe-free focus montages (*Figure 3F*). In our analysis below, we show that these fringes do not impede target detection by 2DTM, making them primarily an aesthetic issue. We also note that the tiling pattern is visible in the montages (*Figure 3B+E*), which we believe is due to the non-linear behavior of the K3 camera since we can observe these shading artifacts in images of a condensed beam over vacuum (*Figure 3—figure supplement 3*).

The montages show membrane vesicles and granules with highly variable sizes and density. We found that a substantial number of granules, which are characterized by higher density inside the surrounding cytosol (*Bainton et al., 1971*), seemed to contain a membrane-enclosed inclusion with density similar to the surrounding cytosol (*Figure 3—figure supplement 4C*) and could therefore be formed by inward budding of the granule membrane. These granules were 150–300 nm in diameter and the inclusions were 100–200 nm in diameter. Based on these dimensions the granules are either azurophil or specific granules (*Bainton et al., 1971*). To our knowledge, these inclusions have not been described in granulocytes and are further described and discussed below.

## 2DTM of DeCo-LACE data reveals large ribosomal subunit distribution in cellular cross-sections

In our initial attempts of using 2DTM on micrographs acquired with the DeCo-LACE protocol, we did not observe any SNR peaks above threshold using the large subunit of the mouse ribosome (*Figure 3—figure supplement 1A*). We reasoned that the edges of the beam might interfere with motion-correction of the movies as they represent strong low-resolution features that do not move with the sample. When we cropped the movie frames to exclude the beam edges, the estimated amount of motion increased (*Figure 3—figure supplement 1B*), consistent with successful tracking of sample motion. Furthermore, in the motion-corrected average we could identify significant SNR peaks (*Figure 3—figure supplement 1B*), confirming the high sensitivity of 2DTM to the presence of high-resolution signal preserved in the images by the motion correction. To streamline data processing, we implemented a function in unblur to consider only a defined central area of a movie for estimation of sample motion, while still averaging the complete movie frames (*Figure 3—figure supplement 1C*). Using this approach, we motion-corrected all tiles in the eight lamellae and found consistently total motion below 1 Å per frame (*Figure 3—figure supplement 2A*). In some lamellae, we found increased motion in the lamella center, which indicates areas of variable mechanical stability within FIB-milled lamellae. In some micrographs, we also observed that the beam edges gave rise to artifacts in the MIP and numerous false-positive detections at the edge of the illuminated area (*Figure 3—figure supplement 1D*). A similar phenomenon was observed on isolated hot pixels in unilluminated areas. To overcome this issue, we implemented a function in the program unblur to replace dark areas in the micrograph with Gaussian noise (see Materials and methods), with mean and standard deviation matching the illuminated portion of the micrograph (*Figure 3—figure supplement 1D*+E). Together, these pre-processing steps enabled us to perform 2DTM on all tiles of the eight lamellae.

We used the tile positions to calculate the positions of the detected LSUs in the lamellae (*Figure 4A*, *Figure 5A*, *Figure 4—video 1*, *Figure 5—video 1*). Overlaying these positions of the lamella montages reveals the LSU distribution throughout the FIB-milled slices of individual cells. Consistent with prior observations imaging selected views in yeast (*Lucas et al., 2022*), organelles like the nucleus and mitochondria only showed sporadic targets detected with low SNRs, consistent with the estimated false-positive rate of one per tile. For each detected target, we also calculated the Z positions from the individual estimated defocus and defocus offset for each tile. When viewed from the side, the ribosome positions therefore show the slight tilts of the lamellae relative to the microscope frame of

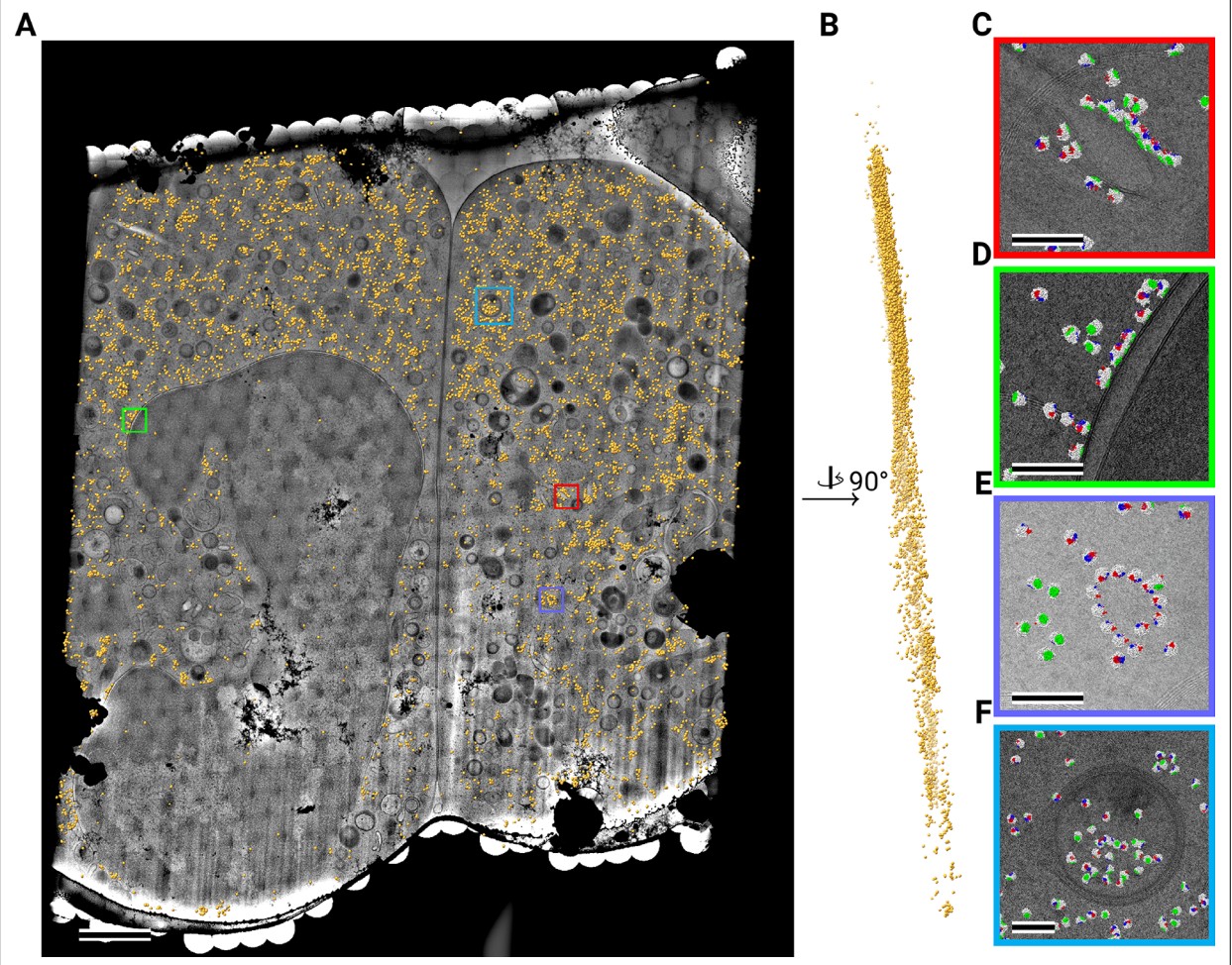

**Figure 4.** Template matching in a lamella imaged using the DeCo-LACE approach at eucentric focus. (**A**) Montage of Lamella$_{EUC}$1 overlaid with detected targets colored in orange. Scalebar corresponds to 1 µm. (**B**) Side view of detected targets in the lamella, such that the direction of the electron beam is horizontal. (**C–F**) Magnified area of panel A showing rough ER with associated ribosomes (**C**), outer nuclear membrane with associated ribosomes (**D**), ribosomes arranged in a circular fashion (**E**), ribosomes enclosed in a less dense inclusion in a granule (**F**). Ribosomes are colored in white with the surface of the peptide exit tunnel colored in green and the A, P, and E sites colored in blue, purple, and red, respectively. Scalebar corresponds to 100 nm.

The online version of this article includes the following video for figure 4:

**Figure 4—video 1.** Movie of detected LSU targets in Lamella$_{EUC}$1, corresponding to Figure 4.
https://elifesciences.org/articles/80980/figures#fig4video1

reference (*Figure 4B*, *Figure 5B*, *Figure 4—video 1*, *Figure 5—video 1*). Furthermore, the side views indicated that lamellae were thinner at the leading edge. To confirm this we estimated the ice thickness in individual tiles using the Beer-Lambert law (*Rice et al., 2018*) with 322 nm as coefficient (*Rice et al., 2018*; *Figure 3—figure supplement 2B*). We found that the thickness of the lamellae varied between 100 and 250 nm, with consistently thinner ice at the leading edge even though we prepared the lamellae with the overtilt approach (*Schaffer et al., 2017*). To further confirm the relationship of the range in Z-positions of detections and sample thickness we plotted the range in Z-positions as a function of the estimated sample thickness (*Figure 3—figure supplement 2C*). For most tiles, the apparent thickness of LSU detections was at least 70 nm less than the estimated sample thickness, indicating that no detections were being made within 35 nm of the lamella edge. It is possible that this is caused by sample damage during FIB-milling as similar values for damaged areas in FIB-milled lamellae have been estimated from sub-tomogram averaging (*Berger et al., 2022*). This means that LSU detections across the lamellae can be skewed by a change in thickness and sample damage.

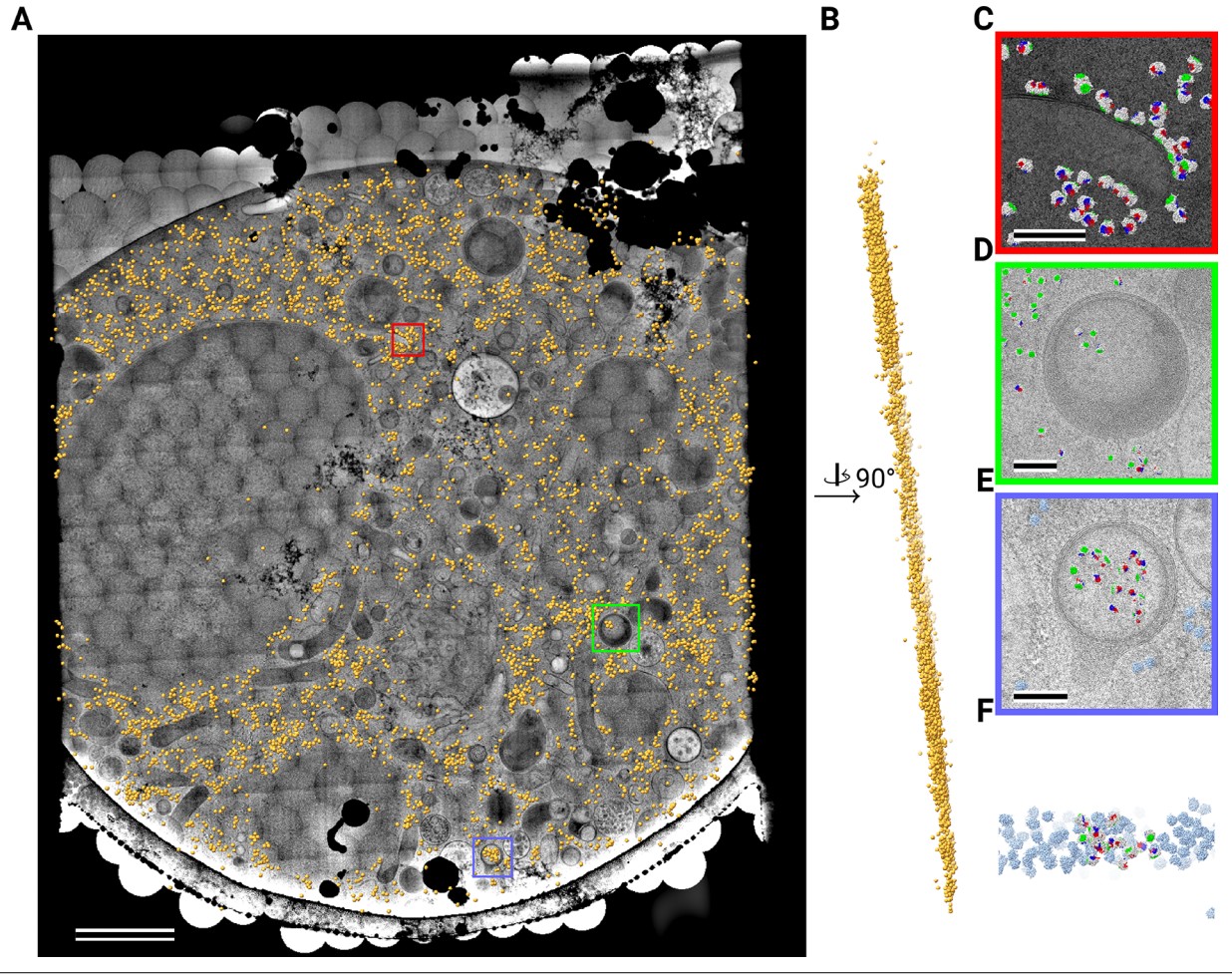

**Figure 5.** Template matching in a lamella imaged using the DeCo-LACE approach at fringe-free focus. (**A**) Montage of Lamella_FFF4 overlaid with detected targets colored in orange. Scalebar corresponds to 1 µm. (**B**) Side view of detected targets in the lamella, such that the direction of the electron beam is horizontal. (**C–E**) Magnified area of panel A showing rough ER with associated ribosomes (**C**) and ribosomes enclosed in a less dense inclusion in a granule (**D,E**). (**F**) Side view of panel E. Ribosomes are colored in white with the surface of the peptide exit tunnel colored in green and the A, P, and E sites colored in blue, purple, and red, respectively. Scalebar corresponds to 100 nm.

The online version of this article includes the following video for figure 5:

**Figure 5—video 1.** Movie of detected LSU targets in Lamella_FFF 4, corresponding to Figure 5.

https://elifesciences.org/articles/80980/figures#fig5video1

Therefore, better sample preparation methods together with a better understanding of the damage to the sample during FIB-milling are needed to accurately quantify molecular concentrations.

As described in *Rickgauer et al., 2017* the 2DTM SNR threshold for detecting a target is chosen to result in one false positive detection per image searched. We would therefore expect to find one false positive detection per tile. We reasoned that the large nuclear area imaged by DeCo-LACE could be used to test whether this assumption is true. In the 670 tiles containing exclusively nucleus (as manually annotated from the overview image) we detected 247 targets, making the false-positive rate more than twofold lower than expected. Since earlier work shows that 2DTM with the LSU can produce matches to nuclear ribosome biogenesis intermediates (*Lucas et al., 2022*), this could even be an overestimate of the false-positive rate. This suggests that the detection threshold could be even lower, which is an area of ongoing research.

Close inspection of the LSU positions in the lamellae revealed several interesting features. LSUs could be seen associating with membranes, in patterns reminiscent of the rough endoplasmic reticulum (*Figure 4C*, *Figure 5C*) or the outer nuclear membrane (*Figure 4D*). We also observed LSUs

forming ring-like structures (*Figure 4E*), potentially indicating circularized mRNAs (*Wells et al., 1998*). While ribosomes were for the most part excluded from the numerous granules observed in the cytoplasm, in some cases we observed clusters of LSUs in the inclusions of double-membraned granules described earlier (*Figure 4F*, *Figure 5D and E*). It is, in principle, possible that these targets are situated above or below the imaged granules, since the granule positions in z cannot be determined using 2D projections. However, in the case of *Figure 5E*, the detected LSUs span the whole lamella in the z direction (*Figure 5F*), while positions above or below a granule would result in LSUs situated exclusively at the top or bottom of the lamella. This is consistent with the earlier hypothesis that the inclusions are of cytoplasmic origin.

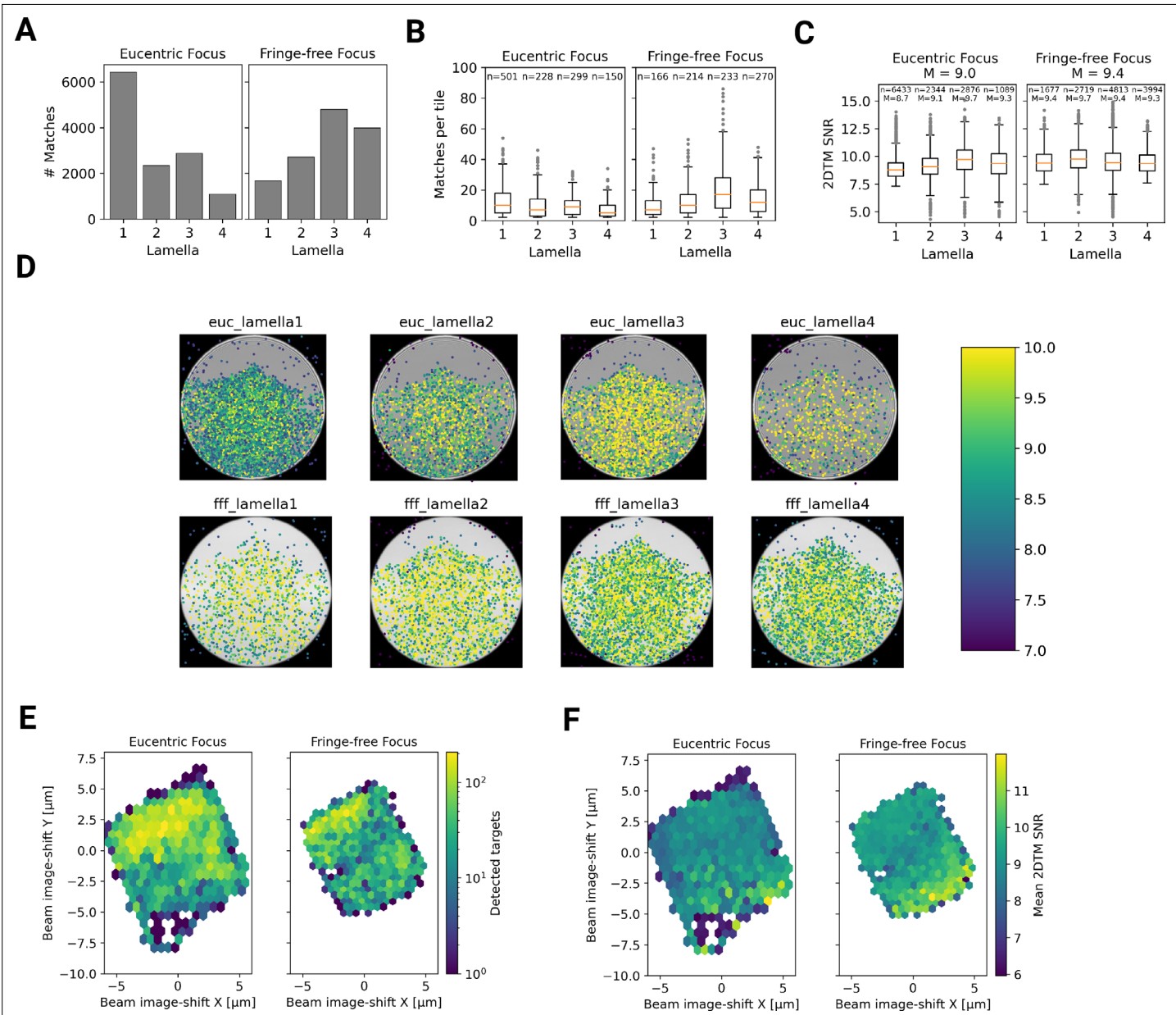

**Figure 6.** Statistics of 2DTM on a lamella imaged using DeCo-LACE. (**A**) Number of detected targets in each lamella. (**B**) Boxplot of detections per tile in each lamella. Only tiles with two or more detected targets were included. Sample size n is indicated for each lamella. (**C**) Boxplot of SNRs in each lamella. Sample size n and median M are indicated for each lamella. (**D**) For each lamella, an average of all tiles is shown. Overlaid is a scatterplot of all detected targets in these tiles according to their in-tile coordinates. The scatterplot is colored according to the 2DTM SNR. There are no detected targets in the top circle-circle intersection due to radiation damage from previous exposures. (**E**) 2D histogram of the number of detected targets as a function of beam-image shift. (**F**) Mean 2DTM SNR as a function of beam-image shift.

## Does DeCo-LACE induce aberrations that affect 2DTM?

Within the eight lamellae we found different numbers of detected targets, ranging from 1089–6,433 per lamella (*Figure 6A*). Lamella$_{EUC}$ 1 had the most detected targets, but also had the largest surface area and contained cytoplasm from two cells. Lamella$_{FFF}$ 4 had the fewest detected targets, but this particular lamella was dominated by a circular section of the nucleus, with only small pockets of cytoplasm (*Figure 3—figure supplement 4*). In an attempt to normalize for these differences in area containing cytoplasm, we compared the number of detected targets per tile in tiles that contained more than one target, which should exclude tiles with non-cytosolic content (*Figure 6B*). While this measure had less variability, there were still differences. Lamella$_{EUC}$ 4 had not only the fewest targets, but also the lowest density, which could be due to this lamella being the thinnest, or due to it sectioning the cell in an area with a lower concentration of ribosomes. Lamella$_{FFF}$ 3 had a substantially higher number of ribosomes per tile. Since all of these lamellae were made from a cell-line under identical conditions, this underscores the necessity to collect data from large numbers of lamellae to overcome the inherent variability. When comparing the distribution of scores between lamellae, we found them to be fairly comparable with median SNRs ranging from 8.7 to 9.7 (*Figure 6C*). Lamella$_{EUC}$ 1 had slightly lower scores compared to the rest, potentially due to its large size and connected mechanical instability during imaging. Overall, we did not observe differences in the number or SNR of detected targets between eucentric or fringe-free illumination conditions that were bigger than the observed inter-lamella variability.

Since the SNR values of 2DTM are highly sensitive to image quality, we reasoned we could use them to verify that DeCo-LACE does not introduce a systematic loss of image quality. We considered non-parallel illumination introduced by the unusually condensed beam and uncharacterized aberrations near the beam periphery. When plotting the SNR values of detected targets in all eight lamellae as a function of their location in the tiles, we found uniformly high SNR values throughout the illuminated areas for both eucentric and fringe-free focus illumination, demonstrating that both illumination schemes are suitable for DeCo-LACE (*Figure 6D*).

We also wondered whether large image shifts would lead to aberrations due to astigmatism or beam tilt (*Wu et al., 2019*). We reasoned that if that was the case the number of detected targets should be highest in the center of the lamella where the applied beam image-shift is 0. Instead, we observed that in both eucentric and fringe-free focus conditions more targets were detected at the back edge of the lamella (*Figure 6E*). This may be due to the center of the cell being predominantly occupied by the nucleus, despite its segmentation in neutrophil-like cells. The increase in matches at the back of the lamellae compared to the front can also be explained by the thickness gradient of the lamellae (*Figure 3—figure supplement 2B*, *Figure 4B*, *Figure 5B*). In addition, aberrations would be expected to cause average 2DTM SNRs to be higher when beam-image shift values are small. Instead, we found that SNRs where on average the highest at the front edge of the lamellae (*Figure 6F*), presumably due to the reduced sample thickness. We therefore conclude that factors other that beam image-shift or beam condensation aberrations are limiting 2DTM SNRS, predominantly the thickness of the lamellae.

## Computation is the bottleneck of visual proteomics

All lamellae described above were derived from a clonal cell line under identical condition and thinned with the same parameters. This means that the substantial variability of detected targets between the lamellae must be due to sample preparation variability, including area, thickness, mechanical stability, and location of the section within the cell. We therefore predict that further studies that want to draw quantitative and statistically relevant conclusions about the number and location of molecules under different experimental conditions, will require collection of orders of magnitude more data than in this study to gain enough statistical power given this variability. The samples used were prepared in two 24 hr sessions on a FIB/SEM instrument, and imaging was performed during another two 24 hr session on the TEM microscope. Inspections of the timestamps of the raw data files revealed that the milling time per lamella was ~30 min and TEM imaging was accomplished in ~10 s per tile or 90 min for a~6 × 6 µm lamella. Processing of the data, however, took substantially longer. Specifically, 2DTM of all tiles took approximately one week per lamella on 32 Nvidia A6000 GPUs. Computation is therefore a bottleneck in our current workflow, and further optimizations of the algorithm may be necessary to increase throughput. Alternatively, this bottleneck could be reduced by increasing the number of processing units.

## Discussion

In this study we developed an approach to image entire cellular cross-section using cryo-EM at high enough resolution to allow for 2DTM detection of the LSU. The two main advantages compared to previous approaches are high throughput of imaging and biological context for detected molecules. The requirement to increase throughput in cryo-EM data collection of cellular samples has been recognized in the recent literature. Most approaches described so far are tailored towards tomography. *Peck et al., 2022* and *Yang et al., 2022* developed approaches to increase the FOV of tomogram data-collection by using a montaging technique. Peck et al. used a similar condensed-beam approach as described here. However, the montages are substantially smaller in scope, covering carbon film holes of 2 m diameter. *Bouvette et al., 2021* and *Eisenstein et al., 2022* are using beam image-shift to collect tilt-series in multiple locations in parallel to increase throughput. However, none of these approaches provide the full coverage of a cellular cross-section that can be achieved using DeCo-LACE.

We observed granules containing a vesicle of putative cytosolic origin. We speculate that upon degranulation, the process in which granules fuse with the plasma membrane, these vesicles would be released into the extracelullar space. The main types of extracellular vesicles of this size are exosomes, up to 100 nm large vesicles derived from fusion of multivesicular bodies with the plasma membrane, and microvesicles, which are derived from direct budding of the plasma membrane (*van Niel et al., 2018*). We suggest that granulocytes could release a third type of extracellular vesicle, granule-derived vesicles (GDV), into the extracellular space. 2DTM showed that a subset of GDVs can contain ribosomes (*Figure 4F*, *Figure 5D and E*). This could indicate that these vesicles are transporting translation-capable mRNAs, as has been described for exosomes (*Valadi et al., 2007*). Further studies will be necessary to confirm the existence of GDVs in granulocytes isolated from mammals and to understand their functional significance.

As mentioned in the results, we found a consistent shading artifact pattern in our montages, that we believe is the result of non-linear behavior of the K3 camera. Indeed, when we average images with a condensed beam taken over vacuum we found in both focus conditions a consistent background pattern with a brighter region on the periphery of the illuminated area (*Figure 3—figure supplement 3*). This might be caused by dynamic adjustment of the internal camera counting threshold which expects columns of the sensor to be evenly illuminated as is the case for SPA applications. Since the signal of this pattern has mainly low-resolution components it is unlikely to affect 2DTM. However, it highlights that the non-linear behavior of the camera has to be taken into account when imaging samples with strongly varying density and unusual illumination schemes.

We found that even though we used beam image-shift extensively (up to 7 μm), we did not see substantially reduced 2DTM SNR values in tiles acquired at high beam image-shift compared to tiles acquired with low or no beam image-shift. This is unexpected because beam image-shift is thought to degrade the high-resolution signal due to the introduction of beam-tilt, as has been observed in single particle analysis (*Cash et al., 2020*; *Cheng et al., 2018*). The signal detected by 2DTM is target and sample dependent and has been shown to be sensitive to a resolution up to 3 Å, but in practice is limited by the B-factor applied to the template (*Rickgauer et al., 2017*). In our case, we used a structure of the mouse LSU that was deposited into the PDB with atoms having a median B-factor of ~80 $Å^2$, which we multiplied by 1.5 before generating the density map. This value was chosen empirically early in the study to optimize the number of detections. The resulting median B-factor of 120 $Å^2$ is equivalent to a Gaussian lowpass filter with a filter constant of 7.7 Å. We did not measure the beam-tilt caused by beam-image shift on the microscope to acquire our data. However, Cheng et. al. report that in their measurements Krios Titan columns exhibited a beam-tilt of 0.17–0.25 mrad per μm of beam image-shift (*Cheng et al., 2018*). We therefore expect a worst-case beam-tilt of 1.75 mrad. At this beam-tilt we expect a phase error of larger than π/4 at resolutions higher than 5.75 Å (*Cheng et al., 2018*), which exceeds the effective resolution of our 2DTM template. We conclude that the beam-tilt caused by beam-image shift did not strongly affect detections in the present study, but it might be important to consider when imaging at even higher beam-image shift values or when using higher resolution signal in 2DTM.

We used 2DTM of the ribosome LSU to verify the image quality of DeCo-LACE data. However, 2DTM has the potential to detect many other protein complexes. The model of the ribosomal LSU used in this manuscript has a molecular weight of ~2 MDa, ~1.3 MDa of which is nucleic acid. The

higher scattering density of nucleic acid compared to protein (*Spahn et al., 2000*) may mean that a biomolecular complex of equal size consisting of only protein would result in lower SNR values. Many biological complexes are smaller and do not contain nucleic acid, but simulations of 2DTM (*Rickgauer et al., 2017*) suggest that it should be possible to detect protein molecules as small as 300 kDa in an crowded environment. Besides the molecular weight we expect that the conformational homogeneity of a protein complex will affect the ability to detect it and note that for complexes that occur at low concentrations inside the cell a higher detection threshold might be necessary to reduce the number of false positives among the few true targets that can be detected.

Since we observed substantial variation in LSU density within and between lamellae, visual proteomics studies that use cryo-EM to establish changes in molecular organization within cells will require orders of magnitude more data than used in this study. One milestone would be to image enough data to represent one cellular volume, which for a small eukaryotic cells requires imaging approximately 100 lamellae. While data collection throughput on the TEM is fundamentally limited by the exposure time, this amount of data could be collected within 12 hr by improving the data acquisition scheme to perform all necessary calculations in parallel with actual exposure of the camera. Sample preparation using a FIB/SEM is also currently a bottleneck, but preparation of large lamellae with multiple cellular cross-sections using methods like WAFFLE (*Kelley et al., 2022*) might allow sufficient throughput. As stated in the results, at least for 2DTM computation will remain challenging and approximately 17,000 GPU hours would be required for a 100 lamellae dataset. It might be possible to increase the performance of 2DTM by improving the implementation of GPU-based fast Fourier transforms. Alternatively, strategies that initially search images at lower resolution and sparser rotational sampling, followed by refinement at higher resolution, could be more efficient, but it is yet unclear whether this is feasible in cellular samples with strong low-resolution background.

## Materials and methods

**Key resources table**

| Reagent type (species) or resource | Designation | Source or reference | Identifiers | Additional information |
|---|---|---|---|---|
| Cell line (*Mus musculus*) | ER-HoxB8 | *Sykes et al., 2016* | | Generated from mouse bone marrow as described |
| Peptide, recombinant protein | SCF | *Sykes et al., 2016* | | Generated from CHO-SCF cell line as described |
| Chemical compound, drug | Estrogen | Sigma | E8875 | |

### Grid preparation

ER-HoxB8 cells were generated as described by Wang et. al. (*Sykes et al., 2016*; *Wang et al., 2006*) and maintained in RPMI medium supplemented with 10% FBS, penicillin/streptomycin, SCF, and estrogen at 37 °C and 5% CO2. 120 hr prior to grid preparation, cells were washed twice in PBS and cultured in the same medium except without estrogen. The authenticity of the cells was verified by flow-cytometry and Giemsa staining after differentiation (*Sykes et al., 2016*; *Wang et al., 2006*). Cells were then counted and diluted to $1 \cdot 10^6$ cells/ml. Grids (either 200 mesh copper grids, with a sillicone-oxide and 2 μm holes with a 2 μm spacing or 200 mesh gold grids with a thin gold film and 2 μm holes in 2 μm spacing) were glow-discharged from both sides using a 15 mA for 45 s. 3.5 l of cell suspension was added to grids on the thin-film side and grids were blotted from the back side using a GP2 cryoplunger (Leica, Wetzlar, Germany) for 8 s and rapidly plunged into liquid ethane at –185 °C.

### FIB-milling

Grids were loaded into an Aquilos 2 FIB/SEM (Thermo Fisher, Waltham, MA) instrument with a stage cooled to –190 °C. Grids were sputter-coated with platinum for 15 s at 45 mA and then coated with a layer of platinum-precursor by opening the GIS-valve for 45 s. An overview of the grid was created by montaging SEM images and isolated cells at the center of gridsquares were selected for FIB-milling. Lamellae were generated automatically using the AutoTEM software (Thermo Fisher), with the following parameters:

- Milling angle: 20°
- Rough milling: 3.2 μm thickness, 0.5 nA current
- Medium milling: 1.8 μm thickness, 0.3 nA current, 1.0° overtilt

- Fine milling: 1.0 μm tchickness, 0.1 nA current, 0.5° overtilt
- Finer milling: 700 nm thickness, 0.1 nA curent, 0.2° overtilt
- Polish 1: 450 nm thickness, 50 pA current
- Polish 2: 200 nm thickness, 30 pA current

This resulted in 6–10 μm wide lamella with 150–250 nm thickness as determined by FIB-imaging of the lamella edges.

## Data collection

Grids were loaded into a Titan Krios TEM (Thermo Fisher) operated at 300 keV and equipped with a BioQuantum energy filter (Gatan, Pleasanton, CA) and K3 camera (Gatan). The microscope was aligned using a cross-grating grid on the stage. Prior to each session, we carefully performed the Image/Beam calibration in nanoprobe. We set the magnification to a pixel size of 1.76 Å and condensed the beam to ~900 nm diameter, resulting in the beam being completely visible on the camera. To establish fringe-free conditions, the Fine eucentric procedure of SerialEM (*Mastronarde, 2005*) was used to move a square of the cross-grating grid to the eucentric position of the microscope. The effective defocus was then set to 2 μm, using the autofocus routine of SerialEM. The objective focus of the microscope was changed until no fringes were visible. The stage was then moved in Z until images had an apparent defocus of 2 μm. The difference in stage Z-position between the eucentric and fringe-free conditions was used to move other areas into fringe-free condition.

Low magnification montages were used to find lamellae and lamellae that were sufficiently thin and free of contamination were selected for automated data collection. Overview images of each lamella were taken at ×2250 magnification (38 Å pixel size). The corners of the lamella in the overview image were manually annotated in SerialEM and translated into beam image-shift values using SerialEMs calibration. A hexagonal pattern of beam image-shift positions was calculated that covered the area between the four corners in a serpentine way, with a $\sqrt{3} \cdot 0.95 \cdot r$ horizontal spacing and $3/2 \cdot 0.95 \cdot r$ vertical spacing, where $r$ is the radius of the illuminated area. For the fringe-free focus condition $r$ was 250 nm and for the eucentric focus $r$ was 230 nm. This resulted in a fractional overlap of adjacent tiles of 10.3% for the fringe-free focus condition and 12.9% for the eucentric focus condition. Exposures were taken at each position with a 30 e⁻/Å² total dose. After each exposure, the defocus was estimated using the ctffind function of SerialEM and the focus for the next exposure was corrected by the difference between the estimated defocus and the desired defocus of 800 nm. Furthermore, after each exposure the deviation of the beam from the center of the camera was measured and corrected using the CenterBeamFromImage command of SerialEM.

After data collection, a 20 s exposure at ×2250 magnification of the lamella at 200 μm defocus was taken for visualization purposes. A Python script implementing this procedure is available at https://github.com/jojoelfe/deco_lace_template_matching_manuscript; *Elferich, 2022b*.

## DeCo-LACE data processing

An overview of the data analysis pipeline is shown in *Figure 7*.

### Pre-processing

Motion-correction, dose weighting and other preprocessing as detailed below was performed using *cis*TEM (*Grant et al., 2018*). To avoid influence of the beam-edge on motion-correction, only a quarter of the movie in the center of the camera was considered for calculation of the estimated motion. After movie frames were aligned and summed, a mask for the illuminated area was calculated by lowpass filtering the image with a 100 Å resolution cutoff, thresholding the image at 10% of the maximal value and then lowpass filtering the mask again with a 100 Å resolution cutoff to smooth the mask edges. This mask was then used to select dark areas in the image and fill the pixels with Gaussian noise, with the same mean and standard deviation as the illuminated area. A custom version of the unblur program (*Grant and Grigorieff, 2015*) implementing this procedure is available at https://github.com/jojoelfe/cisTEM/tree/2574dbdf6161658fd177660b3a841100a792f61b. During motion correction, images were resampled to a pixel size of 1.5 Å. The contrast-transfer function (CTF) was estimated using ctffind (*Rohou and Grigorieff, 2015*), searching between 0.2 and 2 μm defocus.

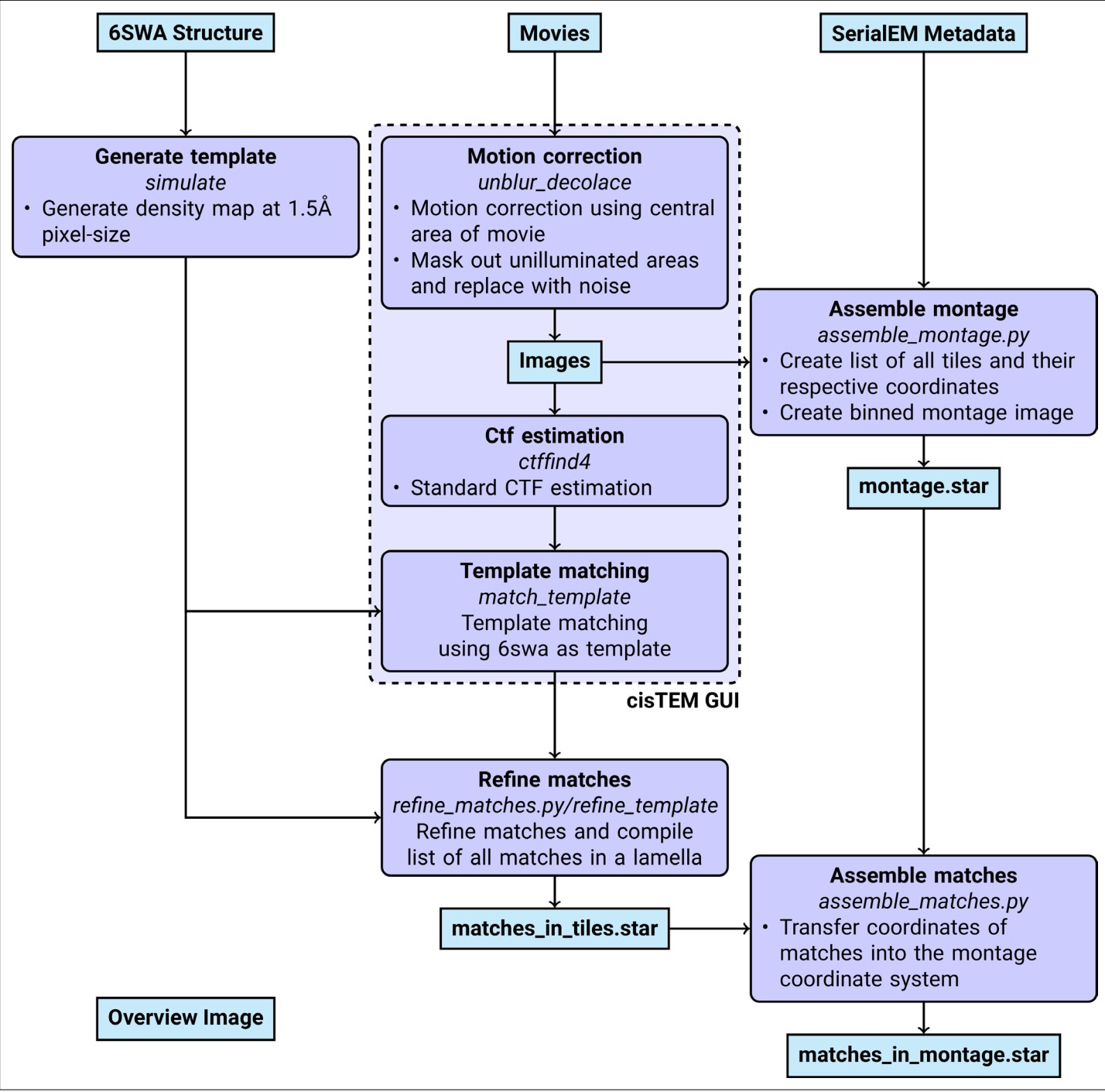

**Figure 7.** Workflow of DeCo-LACE processing.

## 2DTM

The search template was generated from the atomic model of the mouse LSU (PDB 6SWA, exluding the Epb1 subunit) using the cryo-EM simulator implemented in *cis*TEM (***Himes and Grigorieff, 2021***). The match_template program (***Lucas et al., 2021***) was used to search for this template in the movie-aligned, exposure-filtered and masked images, using a 1.5° angular step in out-of-plane angles and a 1.0° angular step in-plane. 11 defocus planes in 20 nm steps centered around the ctffind-determined defocus were searched. Targets were defined as detected when their matches with the template

produced peaks with a singal-to-noise ratio (SNR) above a threshold of 7.75, which was chosen based on the one-false-positive-per-tile criterion (*Rickgauer et al., 2017*).

## Montage assembly

The coordinates of each tile $i$, $\mathbf{c}_i$ [2D Vector in pixels] were initialized using beam image-shift of the tile, $\mathbf{b}_i$ [2D Vector in μm], and the ISToCamera matrix $\mathbf{IC}$, as calibrated by SerialEM:

$$\mathbf{c}_i = \mathbf{IC} \cdot \mathbf{b}_i$$

A list of tile pairs $i, j$ that overlap were assembled by selecting images where $|\mathbf{c}_i - \mathbf{c}_j| < D_{Beam}$. In order to calculate the precise offset between tiles $i$ and $j$, $\mathbf{r}_{i,j}$, we calculated the cross-correlation between the two tiles, masked to the overlapping illuminated area using the scikit-image package (*van der Walt et al., 2014*) to calculate refined offsets. The coordinates $\mathbf{c}_i$ were then refined by a least-square minimization against $\mathbf{r}_{i,j}$:

$$\min_{\mathbf{c}} \sum_{pairs} (\mathbf{r}_{i,j} - (\mathbf{c}_i - \mathbf{c}_j))^2$$

using the scipy package (*Virtanen et al., 2020*). The masked cross-correlation and the least-square minimization was repeated once more to arrive at the final tile alignment.

The x,y coordinates of target $n$ detected by 2DTM in the tile $i$, $\mathbf{m}_{n,i}^{\mathrm{T}}$, was transformed into the montage frame by adding the coordinate of the tile,

$$\mathbf{m}_n^{\mathrm{M}} = \mathbf{m}_{n,i}^{\mathrm{T}} + \mathbf{c}_i$$

The z coordinate of each target was calculated as the sum of the defocus offset for the target, the estimated defocus of the tile, and the nominal defocus of the microscope when the tile was acquired. Images were rendered using UCSF ChimeraX (*Pettersen et al., 2021*) using a custom extension to render 2DTM results available at https://github.com/jojoelfe/tempest (*Elferich, 2022a*). The Python scripts used for data processing are available under https://github.com/jojoelfe/deco_lace_template_matching_manuscript (*Elferich, 2022b*).

# Acknowledgements

The authors thank Bronwyn Lucas, Carsten Sachse, and Chen Xu for helpful suggestions and careful reading of the manuscript as well as members of the Grigorieff lab for helpful discussions. Data was collected at the UMass Chan Medical School Cryo-EM core with help by Kankang Song, Christna Ouch, and Chen Xu.

# Additional information

### Competing interests
Nikolaus Grigorieff: Reviewing editor, *eLife*. The other authors declare that no competing interests exist.

### Funding

| Funder | Grant reference number | Author |
| --- | --- | --- |
| Howard Hughes Medical Institute | HHMI Investigator | Nikolaus Grigorieff |
| National Heart, Lung, and Blood Institute | P01HL131477 | David T Scadden |
| EMBO | ALTF 743-2018 | Giulia Schiroli |

The funders had no role in study design, data collection and interpretation, or the decision to submit the work for publication.

## Author contributions
Johannes Elferich, Conceptualization, Data curation, Software, Formal analysis, Investigation, Visualization, Methodology, Writing - original draft; Giulia Schiroli, Resources, Writing – review and editing; David T Scadden, Conceptualization, Resources, Supervision; Nikolaus Grigorieff, Conceptualization, Supervision, Writing – review and editing

## Author ORCIDs
Johannes Elferich ⬤ http://orcid.org/0000-0002-9911-706X
Giulia Schiroli ⬤ http://orcid.org/0000-0001-9821-7133
David T Scadden ⬤ http://orcid.org/0000-0002-9812-8728
Nikolaus Grigorieff ⬤ http://orcid.org/0000-0002-1506-909X

## Decision letter and Author response
Decision letter https://doi.org/10.7554/eLife.80980.sa1
Author response https://doi.org/10.7554/eLife.80980.sa2

# Additional files

## Supplementary files
• MDAR checklist

## Data availability
Cryo-EM movies, motion-corrected images and 2DTM results have been deposited in EMPIAR under accession code EMPIAR-11063. The custom cisTEM version is available under https://github.com/jojo-elfe/cisTEM/tree/2574dbdf6161658fd177660b3a841100a792f61b (*Grant et al., 2022*) until features have been integrated into a cisTEM release. The ChimeraX extension for rendering is available under https://github.com/jojoelfe/tempest (copy archived at swh:1:rev:f539deaeda18107b58a3e2a0341b-9b130c1ae18c; *Elferich, 2022a*). This manuscript was prepared using the manubot package (*Himmelstein et al., 2019*). The corresponding repository containing all scripts used for figure generation can be found under https://github.com/jojoelfe/deco_lace_template_matching_manuscript (copy archived at swh:1:rev:7e5b699e99e27e6a19325a0e4d7eeb6cbda31e02; *Elferich, 2022b*).

The following dataset was generated:

| Author(s) | Year | Dataset title | Dataset URL | Database and Identifier |
|---|---|---|---|---|
| Elferich J, Shiroli G, Scadden DT, Grigorieff N | 2022 | Cryo-EM data and 2DTM results of entire sections of differentiated ER-HoxB8 cells | https://www.ebi.ac.uk/empiar/EMPIAR-11063/ | EMPIAR, EMPIAR-11063 |

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
