## [Editor Report]

The work details a valuable method of defocus corrected large area cryo-EM (DeCo-LACE). The data-acquisition approach is highly complementary to the research group's previous work of using high-resolution 2D template-matching (2DTM) to identify macromolecular complexes in dense and heterogeneous cellular specimens. Overall, the data presented shows that DeCo-LACE is a solid approach to locating large ribosomal subunits in FIB-lamella at scale.

---

## [Decision Letter]

**Decision letter after peer review:**

Thank you for submitting your article "Defocus Corrected Large Area Cryo-EM (DeCo-LACE) for Label-Free Detection of Molecules across Entire Cell Sections" for consideration by *eLife*. Your article has been reviewed by 2 peer reviewers, and the evaluation has been overseen by a Reviewing Editor and Richard Aldrich as the Senior Editor. The following individual involved in the review of your submission has agreed to reveal their identity: Thomas G Laughlin (Reviewer #1).

The authors have developed a method aimed at characterizing particles found in wide-view image montages from cellular sections. While it might be seen as an incremental advance over their previous publication on template-matching, enabling users to be able to examine wide fields of view has importance and significance for addressing cell biological questions. There are several limitations to the present study, however, that should be addressed in a revised paper:

(1) They have only used the large ribosomal subunit for this study. They should discuss the MW of this object and the fact that it has a significant fraction of nucleic acid, increasing the contrast. They might speculate how the method would work for objects with lower MW and no nucleic acid content.

(2) While the method is proposed as a faster alternative to cryo-ET, the computational demands seem prohibitive. Can they speculate on how the method could be accelerated (and not just by adding more processors)? Given the computational bottleneck, can they provide recommendations to readers on when the presented approach is appropriate?

(3) The manuscript could potentially be improved by a more thorough explanation of the resolution regime to which the 2DTM signal-to-noise ratio (SNR) values are most sensitive. When would one anticipate aberrations like a coma to become problematic?

(4) Fitting of particle defocus allows detection of the Z-height for each large ribosomal subunit. Could the authors estimate what is the precision of the detection of Z-height in their experiment and how it depends on the thickness of the lamella (˜150…250 nm in this case).

(5) How close to the top or the bottom end of the lamella can a ribosome subunit be detected? Can it be detected when it is partially milled away? Do the molecules in the "middle" of the lamella correspond to higher SNR?

*Reviewer #1 (Recommendations for the authors):*

The manuscript could potentially be improved by a more thorough explanation of the resolution regime to which the 2DTM signal-to-noise ratio (SNR) values are most sensitive. When would one anticipate aberrations like a coma to become problematic?

Lines 156-158: How does the inclusion of the non-illuminated areas, even when filled with Gaussian noise, impact the estimate of the false-positive rate? Could this be the source of the apparent overestimation of the false-positive rate?

Ln 189: "exclusively".

Lines 220-224: Are the authors able to incorporate the effect of coma in the 2DTM routine to see its effect on results or perform simulations on the effect coma has on the SNR values? When would one expect a coma to become limiting/negatively affect the SNR? Is it near/beyond the Nyquist of this data set anyway?

Lines 267-269:

Reference 27 (Cash et al., 2020) detailed a case of substantial beam image shift which resulted in |0-6| mrad of beam tilt (up to 20 μm image shift) and limited the reconstruction to 4.9 Å. In Cheng et al., JSB, 2018 the authors could obtain ~3-3.5 Å reconstructions in light of |1.3| mrad beam tilt (~5-8 μm image shift), which is likely closer to the maximum amount of beam tilt being applied in the presented study.

Lines 270-272:

Can the authors elaborate their explanation on the impact of coma on SPA vs 2DTM? I would have thought that a coma would have less of an impact on SPA data through averaging compensatory directions and more of an impact on 2DTM by making the reference project less similar to the experimental image.

Line 342:

Why were images resampled to 1.5 Å? Was this to include information beyond the physical Nyquist? If so, has this been shown to improve the 2DTM results?

Figure-4 supplement-1:

Panels D and E are not described in the legend. Are micrographs cropped to the illuminated area inscribed before 2DTM or is the entirety of the unilluminated area, filled with Guassian noise, included?

Figure 7:

Remove/replace "electron" in the first box.

*Reviewer #2 (Recommendations for the authors):*

The manuscript is technically excellent and well written. I have several important questions that are still not addressed in the current version of the manuscript.

1. Fitting of particle defocus allows detection of the Z-height for each large ribosomal subunit. Could the authors estimate what is the precision of the detection of Z-height in their experiment and how it depends on the thickness of the lamella (˜150…250 nm in this case).

2. How close to the top or the bottom end of the lamella can a ribosome subunit be detected? Can it be detected when it is partially milled away? Do the molecules in the "middle" of the lamella correspond to higher SNR?

3. As all the research using this approach is performed on ribosomal subunits, in order to make the method more general – it should also work on other macromolecules. Could the authors discuss the potential lower limits for detection in FIB lamella?

---

## [Author Response]

The authors have developed a method aimed at characterizing particles found in wide-view image montages from cellular sections. While it might be seen as an incremental advance over their previous publication on template-matching, enabling users to be able to examine wide fields of view has importance and significance for addressing cell biological questions. There are several limitations to the present study, however, that should be addressed in a revised paper:(1) They have only used the large ribosomal subunit for this study. They should discuss the MW of this object and the fact that it has a significant fraction of nucleic acid, increasing the contrast. They might speculate how the method would work for objects with lower MW and no nucleic acid content.

We added a paragraph to the discussion (lines 295-300). The paragraph includes the MW of the template used to detect the large ribosomal subunit (2MDa) and the fraction of RNA. Regarding the amount of included nucleic acid, we refer to Spahn et al. Structure 2000 to discuss the possibility that nucleic acid contribute more the signal in cryo-EM micrographs compared to protein occupying the same volume. Regarding the question about objects with lower MW, we cite Rickgauer et.al. *eLife* 2017, which contains data regarding the MW of proteins that are expected to be detectable. We note that MW might not only be the only criterion that determines detectability, but that the conformational homogeneity and abundance within the cell also play important roles.

(2) While the method is proposed as a faster alternative to cryo-ET, the computational demands seem prohibitive. Can they speculate on how the method could be accelerated (and not just by adding more processors)? Given the computational bottleneck, can they provide recommendations to readers on when the presented approach is appropriate?

We have expanded the paragraph in the discussion with speculation about approaches to accelerate 2DTM (lines 310-313). It is unclear to us whether cryo-ET of a similar sized area would be computationally more efficient, especially given the complex nature and rapid change in processing pipelines. We note that while 2DTM is computationally demanding, there are few steps and little user interaction is required. For the discussion of the advantages and disadvantages of 2DTM compared to cryo-ET we refer to Lucas et.al. *eLife* 2021.

(3) The manuscript could potentially be improved by a more thorough explanation of the resolution regime to which the 2DTM signal-to-noise ratio (SNR) values are most sensitive. When would one anticipate aberrations like a coma to become problematic?

We have updated the discussion paragraph in lines 279-294 with an explanation of the expected resolution ranges compared with the expected coma aberrations as discussed in Cheng et.al. JSB 2018. In summary, we assume that the resolution range being used for 2DTM is lower than the resolution at which coma aberrations are expected to be problematic. (4) Fitting of particle defocus allows detection of the Z-height for each large ribosomal subunit. Could the authors estimate what is the precision of the detection of Z-height in their experiment and how it depends on the thickness of the lamella (˜150…250 nm in this case).

(4) Fitting of particle defocus allows detection of the Z-height for each large ribosomal subunit. Could the authors estimate what is the precision of the detection of Z-height in their experiment and how it depends on the thickness of the lamella (˜150…250 nm in this case).

The precision of Z-height estimation in 2DTM has been previously investigated by comparing it with results from 3D coordinates from cryo-tomography (Lucas. et al. *eLife* 2021). The median difference in the Z-direction between 2DTM and tomography was found to 59 A, roughly a third of the LSU diameter. While we have not yet sufficient data from 2DTM+tomography to perform such an analysis as a function of sample thickness, we note that Lucas et al. used samples with a thickness ranging from 80-220 nm, so we don’t expect uncertainties to be substantialy higher in our present study.

(5) How close to the top or the bottom end of the lamella can a ribosome subunit be detected? Can it be detected when it is partially milled away? Do the molecules in the "middle" of the lamella correspond to higher SNR?

We cannot detect edges of the lamella using a single exposure and therefore cannot directly measure the distance of a detection from the lamella edge. In order to address this question we have estimated the lamella thickness in every tile using the Beer-Lambert law as described in Rice et al., JSB 2018 and compared it to the range of ribosome detections in the beam direction. This analysis suggests that ribosomes are only detected in a slice that is ~70 nm thinner than the lamella, suggesting that we could not detect ribosome ~35nm from the lamella edge, potentially due to damage during the milling process. Our lab is currently performing a more through investigation of this phenomenon.

Reviewer #1 (Recommendations for the authors):The manuscript could potentially be improved by a more thorough explanation of the resolution regime to which the 2DTM signal-to-noise ratio (SNR) values are most sensitive. When would one anticipate aberrations like a coma to become problematic?Lines 156-158: How does the inclusion of the non-illuminated areas, even when filled with Gaussian noise, impact the estimate of the false-positive rate? Could this be the source of the apparent overestimation of the false-positive rate?

During data processing we do not include matches that occur in the Gaussian noise filled non-illuminated areas.

Ln 189: "exclusively".

Thank you for pointing this out. The typo is fixed.

Lines 220-224: Are the authors able to incorporate the effect of coma in the 2DTM routine to see its effect on results or perform simulations on the effect coma has on the SNR values? When would one expect a coma to become limiting/negatively affect the SNR? Is it near/beyond the Nyquist of this data set anyway?

Please refer to the answer to question 3 above.

Lines 267-269:Reference 27 (Cash et al., 2020) detailed a case of substantial beam image shift which resulted in |0-6| mrad of beam tilt (up to 20 μm image shift) and limited the reconstruction to 4.9 Å. In Cheng et al., JSB, 2018 the authors could obtain ~3-3.5 Å reconstructions in light of |1.3| mrad beam tilt (~5-8 μm image shift), which is likely closer to the maximum amount of beam tilt being applied in the presented study.

Thank you for pointing this out. We updated the discussion to include (Cheng et al., JSB, 2018, as described in the answer to question 3).

Lines 270-272:Can the authors elaborate their explanation on the impact of coma on SPA vs 2DTM? I would have thought that a coma would have less of an impact on SPA data through averaging compensatory directions and more of an impact on 2DTM by making the reference project less similar to the experimental image.

We have removed the discussion of the impact of coma in SPA vs 2DTM, since as elaborated in Cheng 2018 the current theory does apparently not capture the experimental observations and as detailed in the answer to question (3) in our case the signal degradation caused by beamtilt occurs at higher resolutions than used by our reference. We are currently exploring including fitting coma and other aberrations during 2DTM to better understand its impact.

Line 342:Why were images resampled to 1.5 Å? Was this to include information beyond the physical Nyquist? If so, has this been shown to improve the 2DTM results?

The resampling to 1.5 Å was not intended to include information beyond Nyquist. Instead we chose 1.5 Å initially since we were expecting information up to 3 Å to contribute to 2DTM and wanted to slightly lower the magnification compared to the ~ 1.0 Å pixelsize use previously, to reduce the number of tiles that have to be acquired. We then chose the first magnification at our Krios that was closest to 1.5 Å physical pixelsize and binned to 1.5 Å, with the idea that this would simplify combining data from different instruments, which we did not do in this study.

Figure-4 supplement-1:Panels D and E are not described in the legend. Are micrographs cropped to the illuminated area inscribed before 2DTM or is the entirety of the unilluminated area, filled with Guassian noise, included?

We apologize for the omission of the figure legend describing panels D and E. Indeed, the micrographs are cropped to the illuminated area in order to avoid performing unnecessary cross-correlation.

Figure 7:Remove/replace "electron" in the first box.

We have removed “electron” as requested.

Reviewer #2 (Recommendations for the authors):The manuscript is technically excellent and well written. I have several important questions that are still not addressed in the current version of the manuscript.1. Fitting of particle defocus allows detection of the Z-height for each large ribosomal subunit. Could the authors estimate what is the precision of the detection of Z-height in their experiment and how it depends on the thickness of the lamella (˜150…250 nm in this case).

Please refer to our answer to question 4 above.

2. How close to the top or the bottom end of the lamella can a ribosome subunit be detected? Can it be detected when it is partially milled away? Do the molecules in the "middle" of the lamella correspond to higher SNR?

Please refer to our answer to question 5 above.

3. As all the research using this approach is performed on ribosomal subunits, in order to make the method more general – it should also work on other macromolecules. Could the authors discuss the potential lower limits for detection in FIB lamella?

Please refer to our answer to question 1 above.